# The Role of Padel in Improving Physical Fitness and Health Promotion: Progress, Limitations, and Future Perspectives—A Narrative Review

**DOI:** 10.3390/ijerph19116582

**Published:** 2022-05-28

**Authors:** Bernardino J. Sánchez-Alcaraz, Javier Courel-Ibáñez

**Affiliations:** 1Department of Physical Activity and Sport, Faculty of Sport Sciences, University of Murcia, San Javier, 30720 Murcia, Spain; bjavier.sanchez@um.es; 2Department of Physical Education and Sport, Faculty of Education and Sport Sciences (Melilla Campus), University of Granada, 52005 Melilla, Spain

**Keywords:** racket sports, physical activity, health promotion, sedentary behaviour

## Abstract

Benefits of regular exercise for health are beyond any doubt. However, adherence to regular physical activity is an ongoing challenge. Among the options for exercise engagement, racket sports, and particularly padel, stand as emerging practices for children and adults to have fun, improve physical fitness, and potentially develop motor and cognitive skills. In the last decade, the literature on padel is increasing exponentially. However, there is a need for further experimental research. To design safe and effective sport-base physical activity promotion interventions, it is essential to have a deep understanding of the physical requirements, technical complexity, injury risks, and strength and conditioning programs. To assist researchers to conduct effective padel-based interventions for health, this review summarizes the state-of-the-art evidence about padel, identifies key topics to be addressed in the future, and discusses the potential role of padel as a physical fitness and health promotion strategy. A narrative review is presented, summarizing the results of padel articles from three different databases: Web of Science, Scopus, and Google Scholar. Studies written in Spanish and English were the inclusion criteria. The studies had to be published from 2000 onwards and be original, as well as peer-reviewed.

## 1. Introduction

The new 2020 World Health Organization (WHO) guidelines on physical activity and sedentary behaviour [1] recommend adults to undertake 150–300 min of moderate-intensity or 75–150 min of vigorous-intensity physical activity per week, or an equivalent combination of both. Furthermore, adults should engage in regular muscle-strengthening activities. However, it is especially worrying how modern society has reduced the time and intensity of physical activity performed in the adult population (35–65 years old) [2]. Inadequate physical activity is considered the fourth cause of death worldwide, especially in developed countries [3,4]. This sedentary lifestyle is associated with important diseases, such as cancer incidence, obesity, hypertension, cardiovascular problems, low bone mineral density, and depression [5,6,7]. In economic terms, physical inactivity is attributed to a total cost of € 49.0 bn to healthcare systems worldwide [8]. Therefore, effective strategies to target increased physical activity amongst adults are warranted [9].

The evidence is uncountable when evaluating the physical and mental benefits of physical exercise programs. Lifetime regular exercise contributes to a better quality of life, improving body composition and cardiometabolic and cognitive conditions [10,11], and delaying the onset of over 40 chronic diseases [12,13,14]. From a psychological point of view, exercise has proven positive effects in controlling depression and anxiety, reducing stress, and modulating the perception of pain [15]. Furthermore, exercise has a role in enhancing neurocognitive abilities, such as learning, concentration, memory, inhibitory control, cognitive flexibility, and the processing of information. Likewise, physical exercise aids in optimizing the evolution of neurodegenerative diseases [16]. More specifically, participants in physical activity programs based on racket sports, such as tennis, appear to have improved aerobic fitness, a lower fat percentage, a more favourable lipid profile, reduced risk for developing cardiovascular disease, and improved bone health [9].

Nonetheless, adherence to regular physical activity is an ongoing challenge. To face this problem, experts suggest engaging in exercise through childhood to promote healthy growth and development [17]. Indeed, evidence supports that physically active children are likely to maintain a healthy lifestyle in adulthood [18]. Among the list of options for practicing exercise, racket sports, and particularly padel, stand as emerging practices for both youth and adults to enjoy, improve physical fitness [19,20], and potentially develop motor and cognitive skills [21].

Padel (aka paddle or paddle tennis) is a relatively new racket sport that was invented in South America in the 1970s [22]. The popularity of padel has seen exponential growth over the last decade, becoming one of the most practiced sports in Spain (counting more than four million regular practitioners), with a solid presence in more than 40 countries around the world [23,24]. This sport is practiced in pairs (2 vs. 2) inside an enclosed synthetic glass and metal court (20 × 10 m) [25]. A particular characteristic of padel is the fact that the ball can bounce on the side and back walls [26], resulting in a faster game rhythm, and it is more dynamic (i.e., players hit the ball more frequently) compared to similar racket sports, without increasing physical intensity [27,28]. Beginning padel practice requires no high technical skills or expensive equipment and can be played outdoors, which makes it a powerful tool for health promotion. Because padel is played massively in recreational environments [23], this sport might play an important role in promoting physical habits among youth and adults.

Recent studies suggest that racket sports can be an effective activity for health promotion and enhance leisure-time physical activity in the sedentary population [29,30]. This is critical in today’s world considering that insufficient physical activity has increased the levels of unhealthy body composition, overweight and obesity at epidemic proportions [31]. Accordingly, padel could be an effective strategy to encourage children and adolescents to undertake regular physical activity for optimal health outcomes and limiting sedentary behaviour, particularly recreational screen time [1].

To design safe and effective sport-based physical activity promotion interventions, it is essential to have a deep understanding of the physical requirements and the technical complexity, injury risks, and strength and conditioning programs. To assist researchers in conducting effective padel-based interventions for health, this review summarizes the state-of-the-art evidence about padel, identifies key topics to be addressed in the future, and discusses the potential role of padel as a physical fitness and health promotion strategy. Furthermore, the aim of this study is to identify the potential role of padel as a physical fitness and health promotion strategy.

## 2. Methods

We conducted a literature search on the main electronic databases (Web of Science, Scopus, and Google Scholar) in December 2021. We limited our search to publications indexed as articles, proceedings papers, and reviews written in Spanish or English, published from 2000 onwards, and which were original and peer-reviewed. No sample restrictions related to age, sex, or category were considered. The search used the combinations of the terms “padel,” or “paddle tennis,” and “racket sport,” or “racquet sport.” After screening for abstract and full texts, the literature search resulted in 36 studies to be included in the review. We categorised these studies into five main research topics: match-play demands (Table 1), biomechanics (Table 2), fitness, body composition and anthropometric characteristics (Table 3), and injuries (Table 4).

## 3. Discussion

### 3.1. Match-Play Demands

A summary of the studies exploring match-play demands in padel is shown in Table 1. Based on the previous research about game activity and the energy requirements in professional and non-professional padel competitions, padel could be defined as an intermittent sport characterized by repetitions of a high frequency of actions (4–6 strokes per rally) during points moderate in duration (10–15 s) at low intensities (work:rest ratio = 1:1.13–1:1.41) [32,33,34,35,36]. The movement patterns showed an average distance of 8–12 m per point and 2500–3500 m per game [37], with short-distance sprints in different directions, which were predominately forward and sideward [27]. The average movement velocities oscillate between 0.60 and 1.70 m/s [38]. The physiological demands have also been analysed in padel [39,40,41]. During a match, padel players showed a mean oxygen consumption (VO_2_) of about 40–50% of the VO_2_ max measured in a laboratory test. Furthermore, similar research reported a mean heart rate (HR) of between 140–160 beats/min (70–80% of maximal HR) [41,42]. These values are lower compared to other racket sports, such as tennis [43], squash [44], or badminton [45]. The intensity patterns may change according to players’ performance levels in padel. High-level players have a significantly higher speed of movement and rate of play (shots per second), rally time, number of shots per point, and distance covered in a match [46]. Because exercise intensity accounts for cardiovascular adaptations, future studies should identify optimal padel doses for health benefits.

**Table 1 ijerph-19-06582-t001:** Summary of studies exploring match-play demands in padel.

Study	Sample	Variables
Hoyo-Lara et al., [39]	Advanced players.	Physiological variables: heart rate, lactate, VO_2_max, and rating of perceived exertion
Carrasco et al., [40]	Advanced players.	Game durations, shot distribution, and physiological variables: heart rate, VO_2_max
Priego et al., [27]	Professional players	Player shots and movements
Amieba et al., [47]	Recreational players	Heart rate, VO_2_max, player movements
Castillo-Rodríguez et al., [41]	Advanced and recreational players	Physiological variables (heart rate, lactate, VO_2_max, and rating of perceived exertion) and players’ movements (distance covered)
Courel-Ibáñez et al., [48]	Professional players	Game result, attack effectiveness, players’ location, and serve situation
Torres-Luque et al., [34]	Professional players	Game durations and shot number and distribution
Carbonell et al., [42]	Semi-professional players	Hear rate and time within training zones
Courel-Ibáñez et al., [28]	Professional players	Performance players’ profiles according to technical, spatial, effectiveness, and hand dominance characteristics
García-Benítez et al., [33]	U-16 and U-18 players	Game durations, rest interval time, and number of strokes per rally
Lupo et al., [35]	Professional players	Shot distribution, shot effectiveness, and rally duration
Díaz-García et al., [49]	Professional players	Mental fatigue and psychomotor vigilance (reaction time)

### 3.2. Stroke Analysis

A summary of the studies exploring stroke analysis in padel is shown in Table 2. Despite the fact that biomechanics is a traditional topic in racket sports with implications for training and injury management, there are only a few studies providing data about motion analyses and kinematics in padel [26,50]. This is critical, considering that the existence of three walls on both court sides in padel is a unique characteristic from which particular offensive and defensive behaviours emerged. A recently published study used a computerized motion tracking video system to examine the type of stroke and ball impact position in highly-trained and novice players [50]. Interestingly, it was found that trained players hit the ball in a more backward position (from 11 to 25 cm, compared to novice players) in serve and offensive strokes (volleys, trays, and smashes) but used more forward strokes (from 7 to 32 cm, compared to novice players) in defensive shots (groundstrokes, wall strokes, and lobs). In another recent study, researchers assessed the differences in the kinematics parameters using two types of balls—Head and Dunlop—in the two existing kinds of wall surfaces—concrete and glass—during a padel drive [26]. The results revealed that the ball type showed a greater influence on the linear space and speed parameters, whilst the wall surface had a greater impact on the ball angle and acceleration. In particular, the glass surface showed faster, longer, and more accelerated bounces than a concrete surface. This may provide some advantages, such as more time to react, and higher balls will increase the odds of scoring directly with a smash. Therefore, playing padel on a concrete surface will result in more difficulties than on a glass surface. Moreover, Dunlop balls described faster and longer bounces than Head balls.

One interesting topic, but barely examined so far, is the velocity of execution, with only one previous study about the overhead smash available [51]. Likewise, available data from a pressurometry and biomechanical study of the foot [52], and the use of kinematic analysis to find out the existence of visual signals (pre-cues) before hitting the ball [53], comes from a single previous study.

**Table 2 ijerph-19-06582-t002:** Summary of studies exploring biomechanics in padel.

Study	Sample	Variables
Priego et al., [52]	Advanced players	Pressurometry and biomechanical variables of the players’ foot
Granda et al., [53]	Advanced players	Visual signals (pre-cues) in drive strokes considering three directions
Rivilla et al., [51]	Advanced and recreational players	Smash variables: ball velocity and precision
Gea et al., [26]	Advanced players	Ball rebound kinematics: angle, acceleration, and speed parameters in two wall surfaces: concrete and glass
Sánchez-Alcaraz et al., [50]	Advanced and recreational players	Ball impact positions (i.e., forward or backward of the centre of gravity) in nine-stroke types

### 3.3. Fitness, Biochemistry, and Body Composition Characteristics

A summary of the studies exploring fitness, biochemistry, and body composition characteristics in padel is shown in Table 3. A few studies have examined the fitness levels of professional padel players [54,55], reinforcing the notion that regular, high-level padel practice induces healthy cardiovascular and strength adaptations. Regarding non-professional padel players, male padel players had good levels of cardiorespiratory fitness, upper body power, handgrip strength, speed, and agility. However, the players’ testing dynamic balance showed low values in both posterior and anterior directions [20]. Other studies showed that adult women that practiced padel had a greater physical fitness condition than sedentary, employing better body balance and explosive power, abdominal endurance, and cardiovascular capacity [19]. Furthermore, they obtained a lower waist/hip circumference and thigh skinfold compared to sedentary. Recent investigations showed an increase in the urinary excretion of some metals involved in antioxidant and energetic functions [56] and several changes in the biochemical parameters related to muscle damage and protein catabolism [57] during a padel match, which seems to indicate a rise in organic metabolism.

The fitness characteristics of young, amateur padel players have been examined lately through cross-sectional studies [58,59,60]. Available results concur that regular padel practice seems not to have an impact on adiposity compared with other racket sports. Besides, the upper and lower limbs’ strength values for young padel players were markedly lower than in other racket sports, with a poorer throwing and jumping ability. Regarding padel players’ sports habits, the frequency of play at the recreational level is 2–3 times per week, 60–120 min per session, and maintained throughout the year [61,62]. According to the last WHO physical activity guidelines [1], this volume of regular practice in healthy and adapted conditions might be beneficial for the health of the general population. However, the long-term benefits of regular padel practice for health are yet to be demonstrated.

**Table 3 ijerph-19-06582-t003:** Summary of studies exploring fitness, biochemistry, and body composition in padel players.

Study	Sample	Variables
Bartolomé et al., [56]	Advanced players	Seven urinary trace elements using ICP-MS chromatography before and after a paddle competition match
Courel-Ibáñez et al., [19]	Recreational adult women	Body composition and physical fitness (static balance, upper body muscular strength, lower body muscular strength, abdominal endurance, flexibility, and aerobic endurance)
Courel-Ibáñez et al., [20]	U-18 players	Body composition and physical fitness (agility, change of direction, and upper limb strength)
Sánchez-Alcaraz et al., [59]	U-18 players	Physical conditioning: strength, speed, and agility
Müller et al., [60]	Amateur players	Cardiovascular fitness, agility, vertical jump height, medicine ball throwing distance, and handgrip endurance strength
Courel-Ibáñez et al., [20]	Advanced and recreational players	Body composition, groundstroke padel accuracy, upper body power, handgrip strength, dynamic balance, agility and speed, and cardiovascular fitness
Pradas et al., [63]	Professional players	Haematological and biochemical values were obtained before and after a simulated competitive padel match
Sánchez-Muñoz et al., [55]	Professional players	Body composition, anthropometric characteristic, handgrip strength, lower-boy muscular strength, lumbar isometric strength, and flexibility
Courel et al., [58]	U-18 players	Body composition, change of direction, agility, and overhead and side medicine ball throws
Pradas et al., [64]	Professional players	Cardiovascular fitness, vertical jump height, and upper and lower limb maximum strength test
Pradas et al., [21]	Trained players	Brain health-related miokines

### 3.4. Injuries

A summary of the studies exploring injuries in padel is shown in Table 4. Regular padel practice induces the repetition of unilateral gestures, which can lead to negative adaptations, such as asymmetries or overuse musculoskeletal injuries. The previous studies have collected data about the injury rates in padel players, involving both professional and amateur samples [65,66,67,68,69]. The reports from the surveys revealed that 40 to 70% of players reported sustaining at least one injury after a year of practicing padel [66,67]. The largest study to date analysed 478 injuries and found an injury rate in this population of 2.75 injuries per 1000 h of risk exposure in padel practitioners [65]. Furthermore, the study identified that the most frequent injuries were those which were mild or with mechanisms of intrinsic injury, and most occurred towards the end of any given game or practice. The lower limb was the most frequently injured body area, and the most injured tissue was the muscle-tendinous system. In turn, epicondylitis was the most common pathology. More specifically, another study examined 158 injuries and found an overall predominance of muscle injuries in the lower limbs, a greater number of trunk injuries in men and upper limbs in women, a higher rate of muscular lesions in players older than 35 years, tendinosis in those younger than 35 years, and a higher rate of injuries in lower level players, especially in their tendons and shoulder [66]. In more detail, a recent case-report study presented how to manage a vascular pathology (the Paget-Schroetter syndrome) in padel players [68] caused by the repetition of shoulder movements and potentially influenced by predisposing factors (venous compression by anomalous anatomical structures).

Although an increase in physical activity intensity can produce even greater benefits for the adults’ health, it also exposes the player to a greater risk of injury [68,70]. Some epidemiology studies have shown a high rate of injury in amateur padel players in the elbow, shoulder, knee, ankle, and lower back, especially in adult recreational players [20,57,63]. Parrón et al. [71] attribute this increasing injury index to bad sports practices since less than 30% of the amateur players perform joint mobility exercises, stretching, or specific strength work. Furthermore, Courel et al. [20] found an alarmingly poor dynamic balance (i.e., the star balance test) in the anterior directions of both high- and low-level players (mean < 75 cm) and balance deficits, which may increase injury risk; therefore, it seems advisable to incorporate lower limb strengthening and balance exercises in padel training routines.

**Table 4 ijerph-19-06582-t004:** Summary of studies exploring injuries of padel players.

Study	Sample	Variables
Castillo-Lozano et al., [72]	Junior and senior players	Incidence of musculoskeletal injuries and risk factors (age, play position, and sport level)
Parrón et al., [71]	Recreational players	Body mass index, sport and nutrition habits, and incidence of injuries
Castillo-Lozano, [70]	Senior players	Epidemiology and prevention strategies for musculoskeletal injuries
Castillo-Lozano et al., [73]	Recreational players	Injury epidemiology (location and incidence) through questionnaires
Priego et al., [67]	Recreational players	Injury epidemiology and risk factors (age, sex, padel participation, and equipment)
García-Fernández et al., [65]	Professional and recreational players	Injury epidemiology (location, incidence, and features) through questionnaires
Sánchez-Alcaraz et al., [66]	Recreational players	Injury epidemiology (location, incidence, and features) through questionnaires
Lozano Sánchez et al., [68]	Case report	Paget-Schroetter syndrome

## 4. Future Perspectives

Whereas indeed the number of scientific papers published on padel is continuously rising, particular research topics would require more attention in the future.

### 4.1. Match-Play Demands

Match analysis is by far the most studied topic in padel, counting several papers about players’ technical-tactical performance from different samples. To provide new insightful information, the implementation of sequential analyses should be considered when analysing the competition due to its practical implications in identifying optimal decision-making strategies and determining those game patterns or chained behaviours that may result in higher offensive or defensive usefulness and effectiveness [74].

### 4.2. Stroke Analysis

Considering the lack of studies on this topic, and because biomechanics studies have practical implications for padel training in both formative and professional contexts, there is a need for further research. For instance, the research focused on kinematic assessment might consider the use of wearable devices, such as inertial measurement units (IMU), which have been proven to be a valid alternative for detecting meaningful differences in the angular velocity during tennis groundstrokes in field-based experimentation [75].

### 4.3. Fitness, Body Composition, and Anthropometric Characteristics

Given that some of the studies have identified a potential association between padel practice and better fitness and body composition status, future studies should confirm this relationship by conducting randomized control trials (RCT). In addition, the relationships between fitness status and performance (i.e., are fit players more successful than unfit players?) and injury risks (i.e., are fit players more likely to suffer an injury than unfit players?) are to be explored yet. The use of mediation analysis could be considered for describing, discovering, and testing possible causal relationships; that is, whether the relationship between two variables (e.g., cardiorespiratory fitness status and body composition) is explained by a third intermediate variable (e.g., regular padel practice experience) [76].

Because the health benefits of regular exercise for health are beyond any doubt, and considering the increasing worldwide interest in padel, better knowledge about the potential impact of acute or regular padel practice on physical and mental health outcomes is needed. Recent studies pointed out that amateur padel practicing stimulates a brain health-related biomarker, such as the brain-derived neurotrophic factor (BDNF) in female players [21]. However, whereas some studies found positive results, to date, all the evidence existing on padel is based on cross-sectional and observational studies. Therefore, researchers must put effort into carrying out controlled trials, including exercise padel-based interventions, both in adult and youth samples, to confirm earlier associations identified.

### 4.4. Injuries

Studies examining padel injuries would benefit from adopting the recently proposed standard methods for the recording and reporting of data for injury and illness in sports by the International Olympic Committee (IOC) [77]. In particular, a tennis-specific extension of the partner IOC statement is now available involving the International Tennis Federation Sports Science and Medicine Committee in collaboration with selected external experts [78]. Compared with the IOC consensus statement, the tennis consensus contains tennis-specific information on the injury mechanism, mode of onset, injury classification, injury duration, capturing and reporting exposure, reporting risk, and study population. Furthermore, examining the effectiveness of injury management and return-to-play programs on padel players would be of interest to coaches and practitioners who usually are forced to adopt racket sports programs because of the lack of padel-specific evidence.

### 4.5. Unexplored Topics to Date

There are key topics that urgently require further examination in padel. For instance, according to a recent systematic review [79], there is only one previous study presenting data on nutrition and ergogenic aids, and in this case, the acute effects of caffeine consumption on specific-padel performance tests [80]. Alarmingly, there are no studies to date examining the effectiveness of strength and conditioning programs on padel players. In this sense, the use and validation of available padel-adapted tests to measure players’ change-of-direction ability [58,81] or stroke accuracy performance [82] seems advisable.

## 5. Limitations

This study presents some limitations. First, the investigations included in this narrative review have used cross-sectional designs with a different population (professional players, recreational adults, females, young players, etc.). Also, narrative studies have some methodological and scientific limitations; hence the data should be interpreted with caution. Any conclusions reached are, therefore, subject to the bias of potentially omitting, perhaps inadvertently, significant sections of the literature or for failing to question the validity of the statements presented. Future studies are now required to verify the effects of interventions based on padel practice in the adult population. In doing so, this may provide more robust evidence toward the benefits of this popular sport in physically inactive populations. In addition, research in padel should address the future perspectives, focusing on unexplored topics or conducting high-quality studies based on the existing knowledge, including also qualitative and mixed research.

## 6. Conclusions

With an increase of padel scientific literature in the last decade, future studies should address new challenges to add to the existing body of knowledge. Scientific research has shown that regular padel practice appears to be recommended to improve physical fitness and promote health, both in adults and children. Also, to assist researchers and padel coaches in conducting effective padel-based interventions for health, this study has summarized the physical requirements, technical and tactical parameters, injury risks, and strength and conditioning programs. 

## Data Availability

Not applicable.

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
