# Peer review of "The Role of Padel in Improving Physical Fitness and Health Promotion: Progress, Limitations, and Future Perspectives—A Narrative Review"

_ijerph, 2022, doi:10.3390/ijerph19116582_

Round 1

Reviewer 1 Report

Dear Authors,

In this article, you proposed a review on the role of padel in improving physical fitness and health promotion. The article is of scientific interest, but I have some concerns about the methodology followed.

Introduction

The introduction adequately argues the background;

Methods

Methodology section is too generic. Thus, I suggest to the author to follow and reference appropriate guidelines for writing a literature review (stating literature search, study selection, inclusion and exclusion criteria for the review, risk of bias in studies…)

Results

3.1. Match-play demands. Please focus on game demands: the economy analysis and the Technical-tactical behaviors are interesting but not appropriate for this section.

Line 139. I suggest to change “biomechanics” in “stroke analysis”.  

Line 167 “Fitness, body composition and anthropometric characteristics”. This should be the most important section of the article considering the title. However it doesn’t represent a deep analysis of the current literature. Please report more detailed data.

Line 189. “3.4. Injury rate and management”. There is no mention in the management of injuries.

“As practical consequences, the mechanism of production is explained by the combination of triggers”. Please rephrase

“This volume of regular practice in healthy and adapted conditions would be very positive at a cardiovascular level.” Please provide a reference.

“However, players with low levels of physical condition and strength would increase the risk of suffering injuries.” Please provide a reference.

“Padel physical demands and workload reported like heart rate, speed of movement and distance covered appears to be beneficial for health-promoting purposes” Please provide a reference.

Author Response

Dear Authors,

In this article, you proposed a review on the role of padel in improving physical fitness and health promotion. The article is of scientific interest, but I have some concerns about the methodology followed.

Introduction

The introduction adequately argues the background;

Thank you!

Methods

Methodology section is too generic. Thus, I suggest to the author to follow and reference appropriate guidelines for writing a literature review (stating literature search, study selection, inclusion and exclusion criteria for the review, risk of bias in studies…)

We have enlarged the methods section according to the reviewer's suggestion.

Results

3.1. Match-play demands. Please focus on game demands: the economy analysis and the Technical-tactical behaviors are interesting but not appropriate for this section.

Agree, we have focused on game demands

Line 139. I suggest to change “biomechanics” in “stroke analysis”.  

Agree, changed.

Line 167 “Fitness, body composition and anthropometric characteristics”. This should be the most important section of the article considering the title. However it doesn’t represent a deep analysis of the current literature. Please report more detailed data.

Accordingly, we have reported more details, data and studies to reinforce this section.

Line 189. “3.4. Injury rate and management”. There is no mention in the management of injuries.

Certainly, literature on injury management is lacking. We have renamed that section “Injuries”

“As practical consequences, the mechanism of production is explained by the combination of triggers”. Please rephrase

We have rephrased the line as “In more detail, a recent case-report study presented how to manage a vascular pathology (Paget-Schroetter syndrome) in padel players, caused by the repetition of shoulder movements and potentially influenced by predisposing factors (venous compression by anomalous anatomical structures)”

“This volume of regular practice in healthy and adapted conditions would be very positive at a cardiovascular level.” Please provide a reference.

We have rephrased the line and included a reference: “According to the last WHO physical activity guidelines [1], this volume of regular practice in healthy and adapted conditions might be beneficial for health in general population.”

“However, players with low levels of physical condition and strength would increase the risk of suffering injuries.” Please provide a reference.

We have deleted this line to avoid confusion.

“Padel physical demands and workload reported like heart rate, speed of movement and distance covered appears to be beneficial for health-promoting purposes” Please provide a reference

We have deleted this line to avoid confusion.

This manuscript is a resubmission of an earlier submission. The following is a list of the peer review reports and author responses from that submission.

Round 1

Reviewer 1 Report

This systematic literature review aims to summarize the state-of-the-art evidence about padel, identifies key topics to be addressed in the future and discusses the potential role of padel as a physical fitness and health promotion strategy.

Congratulations to the authors for the topic under study. However, this review study presents some issues that may need to be addressed. In the comments below, some considerations are set out.

Although the title reveals the scope of the present study, it is suggested to clearly identify that this is a literature review study. You can see that there is a wide variety of review study methods. In addition, the title is of vital importance in selecting and reading review reports. It is recommended to identify the report as to the type of review that was conducted, being consistent with the procedures used.

The abstract is organized in a well-structured format. However, this section can also include other aspects in order for readers to establish a more complete understanding of the findings. In this sense, it is suggested to include information regarding the literature search strategy (e.g., specify databases, eligibility criteria).

Overall, the introduction is well organized. A framework of the problem is presented as well as the definition of key concepts within the study area. However, it is suggested that the authors may situate this literature review, clearly and objectively, within the scope of other previously published reviews on the problem under study. It is suggested to indicate the gaps in knowledge that the present report intends to fill, in the context of other works within the same scope. Furthermore, it is suggested that the concept of "sedentary behavior" be clearly defined, so as not to be confused with the concept of "physical inactivity".

The report presented is a narrative literature review. This type of review has some limitations. Any conclusions reached are therefore subject to the bias of potentially omitting, perhaps inadvertently, significant sections of literature or for failing to question the validity of the statements presented. Furthermore, authors may only select literature that supports their worldview, giving credence to a preferred hypothesis (Grant & Booth, 2009).

In order to minimize these inherent limitations, it would be very important to the reader if the authors could clarify what eligibility criteria were used for the inclusion or exclusion of the studies in this review.

In addition, it is also suggested that a literature search strategy be presented. It is suggested that at least one, search strategy used can be documented, at least for one database, so that it can be replicated by other researchers. Searching from any database, even by experienced researchers, can be imperfect. Thus, further research is suggested, to those conducted.

It would be important if the tables in the report could show other complementary data such as: the country from which the research was carried out, the design of the studies, which data collection instruments were used.

In the context of suggestions for further research, it seems that it would be important to suggest qualitative and mixed research.

Author Response

We would like to thank the reviewer for their detailed comments and suggestions on the manuscript. They have identified important areas which required improvement. We have carefully addressed all the reviewers’ suggestions and provided a detailed point-by-point response to each comment. Please find below a point-by-point response to reviewer’s comment. Reviewer's comments are in bold and numbered (R1.1, R1.2…). Authors' responses are numbered (A1.1, A1.2…). We use red colour to response reviewer #1.

Please, find attached the document with the responses.

Reviewer 2 Report

Dear Authors,

In this article, you proposed a review on the role of padel in improving physical fitness and health promotion. The article is of scientific interest, but I have some concerns about the methodology followed.

General

I suggest a deep grammar check.

Introduction

The introduction adequately argues the background; however, the aim of the study is very generic. I suggest focusing on physical fitness and health promotion

Methods

There is a lack of method section.

The section “The trends in padel scientific literature” can be considered a part of the introduction. Thus, I suggest to the author to follow and reference appropriate guidelines for writing a literature review (stating literature search, study selection, inclusion and exclusion criteria for the review, risk of bias in studies…)

Results

In results section you could include the summary of the studies. However, I suggest to follow the aim of the study (the role of padel in improving physical fitness and health promotion).

 Conclusions

“Scientific research has shown that regular padel practice appears to be recommended to improve physical fitness and to promote health, both in adults and children”. This is the most important part of conclusion. Please argue the sentence.

Limitations should be stated in discussion.

Author Response

We would like to thank the reviewer for their detailed comments and suggestions on the manuscript. They have identified important areas which required improvement. We have carefully addressed all the reviewers’ suggestions and provided a detailed point-by-point response to each comment. Please find below a point-by-point response to reviewer’s comment. Reviewer's comments are in bold and numbered (R2.1, R2.2…). Authors' responses are numbered (A2.1, A2.2…). We use green colour to response reviewer #2.

Please, find attached the document with responses.
